# Characterization of the Bacterial Communities in *Cichorium intybus* According to Cultivation and Storage Conditions

**DOI:** 10.3390/microorganisms11061560

**Published:** 2023-06-12

**Authors:** Su-Jin Yum, Heoun-Reoul Lee, Seon Yeong Yu, Dong Woo Seo, Jun Hyeok Kwon, Seung Min Kim, Jong Hun Kim, Hee-Gon Jeong

**Affiliations:** 1Department of Food Science and Technology, College of Agriculture and Life Sciences, Chungnam National University, Daejeon 34134, Republic of Korea; sujin4380@cnu.ac.kr (S.-J.Y.); lupe47@naver.com (H.-R.L.); dbtjsdud324@naver.com (S.Y.Y.); wnsgur636@naver.com (J.H.K.); 2Division of Human Ecology, Korea National Open University, Seoul 03087, Republic of Korea; kisie@knou.ac.kr; 3Department of Food Science and Biotechnology, College of Knowledge-Based Services Engineering, Sungshin Women’s University, Seoul 01133, Republic of Korea

**Keywords:** chicory leaves, phyllosphere microbiota, food-borne illness

## Abstract

Chicory leaves (*Cichorium intybus*) are widely consumed due to their health benefits. They are mainly consumed raw or without adequate washing, which has led to an increase in food-borne illness. This study investigated the taxonomic composition and diversity of chicory leaves collected at different sampling times and sites. The potential pathogenic genera (*Sphingomonas*, *Pseudomonas*, *Pantoea*, *Staphylococcus*, *Escherichia*, and *Bacillus*) were identified on the chicory leaves. We also evaluated the effects of various storage conditions (enterohemorrhagic *E. coli* contamination, washing treatment, and temperature) on the chicory leaves’ microbiota. These results provide an understanding of the microbiota in chicory and could be used to prevent food-borne illnesses.

## 1. Introduction

Fresh produce is grown on the ground and is usually consumed after minimal processing. Produce may become easily contaminated by food-borne pathogens at any stage of the supply chain, thereby posing a high risk of food-borne illnesses [1,2,3]. According to statistics on food-borne illnesses, fresh produce contaminated with bacteria, such as pathogenic *Escherichia coli*, *Listeria monocytogenes*, and *Salmonella* spp., has led to widespread outbreaks [4,5].

Numerous microorganisms, such as food-borne pathogens with rapid metabolic abilities, survive in natural communities, including raw foods. Food-borne illnesses usually occur in foods contaminated during transport and storage processes and are also caused by pathogens in the indigenous microbiota of fresh produce [6]. In such cases, determining the cause of food-borne illnesses is complicated. In addition, most microbes interact with each other and their habitats, and few microbes live in single-species communities. Understanding the structure and diversity of microbial communities, including potential food-borne pathogens, in raw foods is important for food safety [7]. Although the recent development of next-generation sequencing (NGS) has enabled the identification of food-borne illnesses caused by the indigenous microbiota of environmental samples [8], only a few studies have examined the microbial communities and potential pathogens in fresh produce.

Chicory leaves (*Cichorium intybus*), a perennial alpine plant of the Asteraceae family, are native to Eurasia and can be found in mid-latitudes worldwide [9,10]. Chicory leaves are medically recognized as important plants that are rich in various nutrients, such as inulin, sesquiterpene lactones, cumarins, phenolic acids, vitamins, saponins, and flavonoids [11]. Additionally, chicory leaves have been reported to be effective against jaundice, liver enlargement, gout, and rheumatism. Chicory leaf extracts have been used to treat diabetes, hypolipidemia, and pyorrhea [12]. Meanwhile, the root of commonly dried chicory leaves is used as a coffee substitute, and leaves are used for preparing salads [13]. Consequently, the consumption of fresh or minimally processed chicory leaves has increased recently. Moreover, many researchers have reported on food-borne illnesses caused by chicory leaves; however, compared with chicory roots, little information is available on chicory leaves [14,15].

In this study, we investigated the composition of the phyllosphere microbiota in chicory collected at different times and sites in South Korea. To investigate the risk of potential pathogens, the microbial composition of chicory leaves was compared among different sampling times and sites. We also analyzed the changes in microbiota composition in chicory leaves with respect to the storage conditions after enterohemorrhagic *E. coli* (EHEC) contamination. The results of this study can enhance our understanding of food-borne illnesses caused by the consumption of chicory leaves and help improve management strategies for fresh produce.

## 2. Material and Methods

### 2.1. Sample Collection

The sampling time was selected based on seasons with a maximum production of Chicory (*Cichorium intybus*) in South Korea. Chicory samples were collected during two seasons (spring: March and April, and summer: Jun, July, and September) in 2017 and 2019 from Gongju (36°26′48″ N 127°07′11″ E) and Busan (35°10′46″ N 129°04′32″ E), South Korea. The average temperature in spring was 7.35 ± 5.36 °C (5.39 ± 5.70 °C and 9.32 ± 4.15 °C in Gongju and Busan, respectively). Additionally, the average temperature in Summer was 22.37 ± 5.20 °C (21.05 ± 5.89 °C and 23.69 ± 3.99 °C in Gongju and Busan, respectively). Samples were randomly collected in clean plastic bags and immediately transported to the laboratory under cold conditions using an icebox with ice packs. These samples were immediately placed in a 4 °C refrigerator.

### 2.2. Metagenomic DNA Extraction from Chicory Leaves

Metagenomic DNA was extracted from chicory leaves as described in previous studies [1,2]. Briefly, 25 g of each sample were packed in a sterilized filter bag (FILTRA-BAG; Labplas, Sainte-Julie, QC, Canada) containing 225 mL of Buffered Peptone Water (OXOID, Basingstoke, UK), and homogenized using a BagMixer 400 W (Interscience, Saint-Nom-la-Bretèche, France). After that, samples were centrifuged at 3134× *g* for 10 min at 4 °C, and the supernatants were discarded. The pellet was resuspended in 5 mL of TES buffer (pH 8.0, 10 mM Tris-HCl, 1 mM ethylenediaminetetraacetic acid [EDTA], and 0.1 M NaCl) [16]. The pellets were repeatedly centrifuged twice more to remove impurities (plant tissues) and stored at −80 °C before metagenomic DNA extraction.

The pellets were used for metagenomic DNA extraction, and that was resuspended in 500 μL cetyltrimethylammonium bromide (CTAB; DAEJUNG, Siheung, Republic of Korea) buffer containing 1% polyvinylpyrrolidone (Sigma-Aldrich, St. Louis, MO, USA) and 50 μL lysozyme solution (100 mg/mL). Subsequently, the pellet was incubated at 37 °C for 1 h. After incubation, 200 μL of proteinase K mixture containing 20 μL of 20 mg/mL proteinase K, 140 μL of 0.5 M EDTA, and 40 μL of 10% sodium dodecyl sulfate were added to the pellets and incubated at 56 °C for 1 h. The pellets were centrifuged at 21,206× *g* for 1 min, and supernatant from each tube was transferred to a new microcentrifuge tube. After that, 100 μL of 5 M NaCl solution and 80 μL CTAB/NaCl solution were added into the tube. After vortexing, the bacterial cells were then mixed with an equal volume of phenol/chloroform/isoamyl alcohol (25:24:1 *v*/*v*/*v*), and the samples were mixed and centrifuged at 21,206× *g* for 5 min. The supernatant was transferred to a new tube, and an equal volume of chloroform was added and mixed. The above process was repeated two more times. Thereafter, the sample was centrifuged at 21,206× *g* for 5 min. After transferring the supernatant to a new tube, 3 μL of RNase A (100 mg/mL) were added and incubated at 37 °C for 1 h. Residual RNase A was removed by extraction with the phenol-chloroform process. The DNA mixture included 10% volume of 3 M sodium acetate (pH 5.0), and 2 volumes of ice-cold 100% ethanol were added to the DNA mixture. The sample was centrifuged at 21,055× *g* for 20 min at 4 °C [17], and the supernatant was discarded. Each pellet was washed with 1 mL of 70% ice-cold ethanol. After air drying, DNA pellets were resuspended with 50 μL of TE buffer (pH 8.0, 10 mM Tris-HCl and 0.1 mM EDTA) and incubated at 55 °C for 1 h. The extracted metagenomic DNA was determined using a microplate spectrophotometer (Multiskan GO microplate Spectrophotometer; Thermo Scientific, Santa Clara, CA, USA) and 1.2% agarose gel electrophoresis. The metagenomics DNA was stored at −20 °C.

### 2.3. Bacterial 16S rRNA Gene Amplification and MiSeq Sequencing

The bacterial 16S rRNA gene (V5-V6 region, 382 bp) was amplified using the specific barcoded primers (Appendix A). Barcoded primers were used for NGS with Illumina MiSeq sequencing (Illumina, San Diego, CA, USA) [18,19]. The polymerase chain reaction (PCR) products were purified using the MEGAquick-spin Plus total fragment DNA purification kit (iNtRon, Seoul, Republic of Korea). Index PCR was performed using an Illumina Nextera XT Index kit, and the library was purified using AMPure XP beads (Beckman Coulter Inc., Brea, CA, USA) according to the manufacturer’s instructions. The size and quality of the libraries were assessed using an Agilent Bioanalyzer DNA 1000 chip kit and the KAPA qPCR kit (KAPA Biosystems, Wilmington, MA, USA). Equal amounts of libraries from all samples were pooled, and 300 bp paired-end MiSeq sequencing was conducted using the Illumina MiSeq platform by LabGenomics (Sungnam, Republic of Korea).

### 2.4. Microbiota Profiling and Diversity Analysis

Raw sequencing data were obtained using the MOTHUR software (ver. 1.38.1) to trim (adapter and primer sequences) and remove chimeric sequences (homopolymers > 8 and average quality score < 25). The average length after trimming was between 250–300 bp. Operational taxonomic units (OTUs) were determined using the CLC Microbial Genomics Module on the CLC Genomics Workbench (ver. 9.5.3, CLC Bio, Aarhus, Denmark). The taxonomic arrangement of the non-chimeric reads was determined using the SILVA database (ver. 123) with an 80% confidence threshold. They were clustered based on 99% sequence similarity for the high specificity and sensitivity of infected *Escherichia* on chicory leaves. Prior to analyzing microbial diversity, sequence readings from the plant archaea, chloroplasts (*Streptophyta*), and mitochondria (*Raphanus*) were removed from the OUTs results. The 38,201 validated reads per sample were compared with α-diversity (Observed OTUs, Chao1 index, and Shannon index), β-diversity (Weighted UniFrac and Unweighted UniFrac), and composition of microbiota at several taxonomy levels (Phylum, Family, and Genus). In case any sequence could not be assigned to a sublevel in the classification process, every unknown taxonomy was defined as “Uncultured” (ex: *Enterobacteriaceae*_unc).

### 2.5. Total Bacterial and Potential Pathogenic Bacterial Quantification Using Quantitative Real-Time PCR (qRT-PCR)

Total bacteria and potentially pathogenic species were quantified using qRT-PCR (CFX Connect Optics Module; Bio-Rad, Hercules, CA, USA) to determine bacterial loads. The PCR primers used are listed in Appendix A. PCR reaction mixtures were assembled from 10 μL SSoAdvanced universal SYBR Green supermix (2×; Bio-Rad), 10 μM each primer, and 1 μL metagenome DNA template or distilled water (negative control) in a final volume of 20 μL. PCR cycling was performed at 98 °C for 3 min (preheating), 98 °C for 15 s (denaturation), 56 °C for 30 s (annealing), 72 °C for 30 s (extension), and 72 °C for 5 min (final extension) in the CFX connect real-time PCR Detection system (Bio-Rad). Genomic DNA (gDNA) of pathogenic strains was extracted using the NucleoSpin Microbial DNA kit (Macherey-Nagel, Düren, NRW, Germany). The standard curves were generated using 10-fold dilutions of this bacterial gDNA (10^2^ to 10^6^). The cycle threshold (Ct) values were determined after adjustment of the baseline (1000) using the CFX Manager software (BioRad, Hercules, CA, USA). The bacterial loads were quantified by comparing their Ct value to the standard curve. The regression coefficient (r^2^) value of the standard curve was greater than or equal to 0.99. All DNA samples were analyzed in triplicate.

### 2.6. Artificial Contamination of Enterohemorrhagic E. coli (EHEC)

To investigate the shift in chicory leaves microbiota due to infection of food-borne pathogens during storage, the artificial contamination of EHEC (*E. coli* ATCC 35150) was conducted at various storage conditions, including washed, unwashed, and at two temperatures (4 °C and 26 °C) in triplicate. EHEC was cultured at 36 °C in Luria-Bertani medium and diluted by BPW. Chicory leaves were washed using the Ministry of Food and Drug Safety method (washing for 30 sec after dipping for 5 min in tap water) [2].

Chicory leaf samples (*n* = 56) were infected using spotting methods [20]. BPW was also spotted on the leaves as a non-contaminated group. The 100 μL of EHEC inoculum (1.75 × 10^5^ CFU) were evenly spotted at 10 locations on the surface of chicory leaves. Then, all the chicory leaves were dried for 30 min prior to storage. The metagenomic DNA was extracted as described in the DNA extraction section.

### 2.7. Statistical Analysis

Statistical significance of total bacterial loads and all diversity indices was conducted using the Student’s *t*-test. Results with *p*-values < 0.05 were considered significantly different when analyzed using the SAS program (ver. 9.4; SAS Institute, USA) and Prism (ver. 5.02). A linear discriminant analysis effect size (LEfSe) analysis was performed using the Huttenhower Lab Galaxy server (https://huttenhower.sph.harvard.edu/galaxy, accessed on 31 May 2023). An alpha level of 0.05 and an exceeded LDA log score of ± 2.0 were used as thresholds for significance in the bar plot of LEfSe analysis.

## 3. Results and Discussion

### 3.1. Comparison of Bacterial Loads and Diversity Indices on Chicory Leaves among Sampling Times and Sites

A total of 3,056,043 reads (an average of 38,201 reads) were obtained from 80 chicory leaf samples. The read number was randomly normalized to 18,840 reads (the minimum read number of chicory leaf samples) per sample to compare the diversity indices (Table 1).

The total bacterial count significantly differed according to the sampling time (spring and summer) and site (Gongju and Busan). The number of total bacteria in the spring samples (6.78 log CFU/g) was higher than that in the summer samples (6.11 log CFU/g, *p* < 0.0001). The bacterial loads detected from the Gongju samples (6.76 log CFU/g) were also higher than those in the Busan samples (6.19 log CFU/g, *p* < 0.0001) in both sampling times (spring and summer). The microbial diversity of chicory leaves was evaluated using alpha (observed OTUs, Chao1, and Shannon) diversity indices. Although the Chao1 indices of chicory leaves were not significantly different depending on the sampling time (610.09 in spring and 600.30 in summer, *p* = 0.78) or site (635.13 in Gongju and 575.26 in Busan, *p* = 0.08), the observed OTUs and Shannon index of the spring (379.48 and 4.51, respectively) were significantly lower (*p* < 0.05 and *p* < 0.01, respectively) than those in the summer (431.37 and 5.18). In addition, the observed OTUs and Shannon index of the Gongju samples were significantly higher than those of the Busan samples in spring (*p* < 0.01). Therefore, differences in bacterial load and diversity exist according to the sampling time and site, and it is expected that dominant bacteria exist in spring chicory samples. In previous reports, bacterial load and diversity were found to be influenced by various environmental conditions such as temperature, humidity, soil conditions, cultivation method, and interactions among microbes, depending on the sampling time and site [21,22]. Our results showed that the microbiota of chicory leaves differed depending on the environmental cultivation conditions.

### 3.2. Microbiota Composition of Chicory Leaves by Principal Coordinate Analysis (PCoA)

We analyzed the microbiota using PCoA based on unweighted and weighted UniFrac distance matrices (Figure 1). Unweighted and weighted PCoA plots of chicory leaves indicated a clear clustering pattern according to sampling time (spring and summer). However, chicory leaf microbiota did not differ between the Gongju and Busan samples. These results indicate that temporal factors (temperature and humidity) influence the microbiota on chicory leaves to a greater extent than site factors (soil conditions and cultivation methods). Furthermore, previous studies on fresh produce microbiota have reported many temporal factors that can affect these microbial clusters [1,23].

### 3.3. Comparison of Microbiota at the Phylum, Order, Family, and Genus Levels

The bacterial community composition of the chicory leaves was compared at the phylum, family, and genus levels. Proteobacteria, Firmicutes, and Actinobacteria were the dominant phyla (more than 2% on average) in chicory leaves (Figure 2). In spring, Proteobacteria (94.20 ± 7.42%, *p* < 0.001) showed higher abundance, whereas Firmicutes (3.25 ± 5.61%, *p* < 0.0001) showed lower abundance. The abundance of Actinobacteria was not significantly different between the spring (2.57 ± 3.07%) and summer (2.74 ± 3.36%, *p* = 0.11). These differences in microbiota composition may be attributed to cultural and environmental factors. Proteobacteria, Firmicutes, and Actinobacteria were also the dominant phyla in plant phyllospheres, such as berries, Chinese chives, spinach, cabbage, perilla, grape leaves, and lettuce, and their abundance differed depending on the sampling time [2,23,24].

At the order level, Enterobacteriales, Pseudomonadales, Rhizobiales, Sphingomonadales, Bacillales, Burkholderiales, and Oceanospirillales were dominant (over an average of 3%). Pseudomonadales showed higher abundance in spring (50.08 ± 22.11%) than in summer (7.89 ± 7.19%, *p* < 0.0001). The abundance of Enterobacteriales, Bacillales, and Oceanospirillales was lower in spring (16.63 ± 12.51, 2.96 ± 5.59, and 0.52 ± 3.00%) than in summer (42.88 ± 29.32, 8.02 ± 11.00, and 6.50 ± 13.82%, *p* < 0.01). In contrast, Rhizobiales, Sphingomonadales, and Burkholderiales did not differ significantly between the sampling times. These taxa contain various potential pathogens and have been reported as indigenous microbiota in various environments, including fresh produce [25,26].

At the family level, *Enterobacteriaceae*, *Pseudomonadaceae*, *Methylobacteriaceae*, *Sphingomonadaceae*, and *Rhizobiaceae* (included in the phylum Proteobacteria) accounted for an average of more than 0.5% of chicory leaves. *Enterobacteriaceae* were more dominant in summer (42.88 ± 29.32%) than in spring (16.63 ± 12.51%). *Pseudomonadaceae* were more dominant in spring (48.90 ± 22.72%) than in summer (6.81 ± 6.40%). *Methylobacteriaceae* (5.92 ± 5.69% in spring and 9.58 ± 14.32% in summer), *Sphingomonadaceae* (6.69 ± 5.03% in spring and 5.16 ± 7.81% in summer), and *Rhizobiaceae* (5.82 ± 6.76% in spring and 4.65 ± 6.73% in summer) did not significantly differ between spring and summer. In previous studies, *Enterobacteriaceae* and *Pseudomonadaceae* were dominant in soil, water, and fresh produce (fruits and vegetables) [27,28]. These results are consistent with those of the present study, showing the composition of the chicory leaf microbiota.

Heatmap analysis was used to compare the microbiota of chicory leaves at the genus level (over an average of 0.5%, Figure 3). Differences in the abundance of some pathogenic genera are shown in the three groups. *Methylobacterium* (5.92 ± 5.69% in spring and 9.57 ± 14.32% in summer), *Sphingomonas* (6.69 ± 5.03% in spring and 4.90 ± 7.37% in summer), and *Rhizobium* (5.77 ± 6.76% in spring and 4.53 ± 6.60% in summer) were dominant genera in both sampling times. These genera are widely distributed in the environment, such as soil and water, and have a wide range of growth temperatures [29]. *Pseudomonas* and *Serratia* in spring (48.90 ± 22.72% and 2.96 ± 3.91%) were significantly higher than in the summer samples (6.81 ± 6.40% and 0.57 ± 0.95%, A group). Some species of *Pseudomonas* and *Serratia* have been reported to survive at low temperatures through physiological changes that neutralize the problems caused by low temperatures [30,31,32]. Moreover, these species are related to food-borne pathogens and spoilage bacteria, such as *P. aeruginosa*, *P. fluorescens*, and *S. marcescens*. *Pantoea*, *Enterobacter*, *Escherichia*, *Bacillus*, and *Klebsiella* in spring (5.78 ± 7.85%, 1.02 ± 1.29%, 0.01 ± 0.001%, 0.09 ± 0.11%, and 0.00 ± 0.00%) were significantly lower than that in the summer samples (14.33 ± 14.21%, 10.54 ± 10.40%, 7.27 ± 11.80%, 4.13 ± 7.16%, and 4.13 ± 6.05%, B group). The genera *Pantoea*, *Enterobacter*, *Escherichia*, and *Klebsiella* in the *Enterobacteriaceae* family are found in various environments, such as soil, water, animals, plants, insects, and humans [33]. The genus *Bacillus* is widely distributed in soil and water [34]. These dominant genera appeared to be the endogenous microbiota in chicory leaves derived from the soil and water. These are the indigenous microbiota of chicory leaves containing various potential food-borne pathogens, such as *P. agglomerans*, *E. sakazakii*, *E. coli*, *K. pneumoniae*, and *B. cereus*, which are closely related to food-borne illnesses. These potentially pathogenic genera cause food-borne illnesses because of their dominance in the production and distribution processes. Therefore, care should be taken to prevent food-borne illnesses caused by chicory leaf microbiota during the production and distribution processes, regardless of the sampling conditions.

### 3.4. Quantification of Potential Pathogenic Species on Chicory Leaves

The potential pathogens in the chicory leaf microbiota at the genus level were quantified by qRT-PCR using specific primers (Table 2). Enterohemorrhagic *E. coli* (EHEC), Enteropathogenic *E. coli* (EPEC), and *B. cereus* were only detected in the spring samples (6.59 × 10^2^ CFU/g, 3.28 × 10^2^ CFU/g, and 5.12 × 10^3^ CFU/g; 5.00%, 7.50%, and 2.50%, respectively). Enterotoxigenic *E. coli* (ETEC) was not detected in spring or summer. EHEC is a highly pathogenic subgroup of Shiga toxin-producing *E. coli* (STEC) that causes bloody diarrhea, abdominal cramps, and hemolytic uremic syndrome (HUS) [35]. EHEC serotype O157:H7 is a human pathogen associated with HUS [36]. EPEC remains an important cause of fatal infant diarrhea in developing countries; however, the mechanism that causes diarrhea is unknown [37]. The major virulence factors of *B. cereus* are hemolysin, enterotoxin, and emetic toxin, which confer resistance to antibiotics, heat, and irradiation treatment by spore formation [38]. *K. pneumoniae* (7.36 × 10^2^ CFU/g, 8.75%)*, S. marcescens* (2.57 × 10^3^ CFU/g, 1.25%)*,* and *A. lwoffii* (1.09 × 10^2^ CFU/g, 2.50%) were detected only in summer. *K. pneumoniae* had the highest detection rate of 8.75% (7.36 × 10^2^ CFU/g) among all potential pathogens. The soil bacterium *S. marcescens* is a nosocomial pathogen found in many environmental niches that can infect plants and animals [39]. *K. pneumoniae* is an opportunistic pathogen that accounts for 10% of nosocomial bacterial infections and causes kidney failure, lung infections, and encephalitis [40,41]. *A. lwoffii* can be isolated from sewage, soil, water, and a wide variety of food resistant to the antibiotics such as carbapenem and tetracycline (β-lactams class) [42,43]. The bacterial loads and detection rates in the chicory leaves were higher in spring than in summer. These results differ from those of previous studies, in which food-borne pathogens were detected more frequently in vegetables during summer [1,23]. These results show that the characteristics of chicory leaves, rather than the cultivation conditions, affect pathogen growth. The detection rate of potential pathogens is low at the production stage, but caution is required, as contamination during the distribution process may increase the risk of food-borne illness.

### 3.5. A Shift in Chicory Leaf Microbiota following Artificial Infection at Various Storage Conditions

In this study, our results showed that the *Escherichia* spp., including EHEC, belongs to the indigenous microbiota on chicory leaves. This suggests that indigenous EHEC can cause a risk related to food safety during the storage or transporting of chicory leaves. Therefore, the effects of EHEC artificial inoculation on microbiota under various storage conditions (washing, EHEC contamination, storage temperature, and storage time) were analyzed. A total of 5,252,515 reads (average of 109,427 reads) were obtained from 48 chicory leaf samples. Changes in the indigenous microbiota according to storage conditions were observed via 16S rRNA gene profiling, and bacterial loads were determined (Figure 4).

Although higher temperatures significantly influenced the change in the microbiota structure of chicory leaves, obvious effects on the microbiota due to washing and EHEC contamination were observed (Figure 4A). The abundance of several genera, such as *Bacillus*, *Enterobacter*, *Escherichia*, *Erwinia*, *Methylobacterium*, *Pseudomonas*, *Rhizobium*, *Sphingomonas*, and *Stenotrophomonas*, was significantly influenced by these factors. Among them, the abundance of *Escherichia* was 5.75 ± 2.22% at 0 h, but it increased to 74.25 ± 6.29% (highest abundance among all samples) when the contaminated chicory leaves were stored at 26 °C after washing treatment. The microbiota composition of fresh produce can be changed in response to various postharvest management practices. In a previous study, distinct changes in the microbiota structure of fresh produce were attributed to sanitation practices in unwashed and washed samples [44]. In our results, the pathogenic *Escherichia* genus within the chicory leaves microbiota showed a significant increase when samples were washed and stored at 26 °C. This seems to have occurred as particular microorganisms, after washing, destroyed the microbiota structure, grew rapidly at 26 °C, and became dominant.

To accurately estimate absolute taxon abundance from the NGS data, total bacteria and EHEC were quantified using qRT-PCR. The total bacterial loads of chicory leaves stored at 26 °C were higher than those stored at 4 °C, regardless of the EHEC contamination or washing treatment (Figure 4B,C). However, the total number of bacteria, depending on the washing treatment, was significantly higher in the unwashed group. In addition, significant differences were observed only at 24 h in both the washed and non-washed groups. Considering this together with the previous results shown in Figure 4A, these results indicate that the washing treatment obviously decreased the total bacteria and disrupted the indigenous microbiota structure.

A higher amount of EHEC in the 26 °C group was determined compared to that in the 4 °C group during 24 h storage (Figure 4D). The number of EHEC in the chicory leaves stored at 26 °C increased until 12 h, while that number decreased in the unwashed group stored at 4 °C. In general, most bacteria on fresh produce have been reported to increase significantly when stored in higher temperatures than in refrigeration conditions, which seems to be a reasonable result [45]. Interestingly, statistically higher amounts of EHEC in washed chicory leaves were determined than in unwashed samples after storing for 12 h and 24 h at 4 °C. This result indicates that storing washed chicory leaves at 4 °C strongly disrupts the microbiota structure, allowing potential food-borne pathogens, such as EHEC, to grow longer at lower temperatures.

Taken together, the effect of the washing treatment and storage at 4 °C on microbiota structure was evident. Therefore, using LEfSe analysis, we identified a shift in the indigenous microbiota due to storage conditions without artificial contamination (Figure 5). In general, a higher absolute LDA value indicated that the species were more enriched in the group [46]. We found that *Pantoea* (c), *Serratia* (d), *Pseudomonas* (e; affiliated to the *Pseudomonadaceae* family within the Pseudomonadales order), and *Stenotrophomonas* (h; affiliated to the *Xanthomonadaceae* family within the Xanthomonadales order) at the genus level were significantly enriched in the unwashed group stored at 4 °C (LDA score ≥ 4.0; *p* < 0.05). However, Rhizobiales at the order level (a) and *Escherichia* at the genus level (b) were significantly enriched in the washed group stored at 4 °C with LDA scores of 5.00 and 4.59 (*p* < 0.05). This suggests that the washing treatment may disrupt the indigenous microbiota on chicory leaves, allowing the contaminated EHEC to colonize better and survive at low temperatures.

Consumption of fresh produce is increasing because of its health benefits [47]; however, this can increase the frequency of food-borne illnesses related to fresh produce [48]. In this study, we analyzed the composition of chicory leaf microbiota cultivated in South Korea at different sampling regions and times to detect potential pathogens. This study also showed that the shift in microbiota structure, including potential pathogens, can be affected by different storage conditions, such as temperature and washing treatment. Our results can contribute to preventing the contamination of food-borne pathogens from any specific type of treatment, packaging, or storage of fresh produce.

## 4. Conclusions

The microbiota of chicory leaves is related to temporal factors, such as harvesting time, temperature, and humidity conditions. Proteobacteria, Firmicutes, and Actinobacteria were the dominant phyla in chicory leaves during spring and summer. Enterobacteriales, Pseudomonadales, Rhizobiales, Sphingomonadales, Bacillales, Burkholderiales, and Oceanospirillales were dominant at both time points. The dominant families were Enterobacteriaceae, Pseudomonadaceae, Methylobacteriaceae, Sphingomonadaceae, and Rhizobiaceae. *Pseudomonas* and *Serratia* were the dominant genera in the spring samples, whereas *Pantoea*, *Enterobacter*, *Escherichia*, *Bacillus*, and *Klebsiella* were dominant in the summer samples. The detection rate and number of pathogenic species belonging to the indigenous microbiota of chicory leaves in summer samples were higher than those in the spring samples. We also analyzed the influence of washing treatment and EHEC contamination on the chicory leaf microbiota. Although microbiota composition differed between diverse storage conditions, chicory leaves stored at 4 °C after washing treatment or those stored at 26 °C showed higher risk due to EHEC contamination. Therefore, the microbial composition of chicory leaves determined in this study can be used as basic data for food safety management to prevent food-borne illness outbreaks caused by the consumption of fresh produce.

## Figures and Tables

**Figure 1 microorganisms-11-01560-f001:**
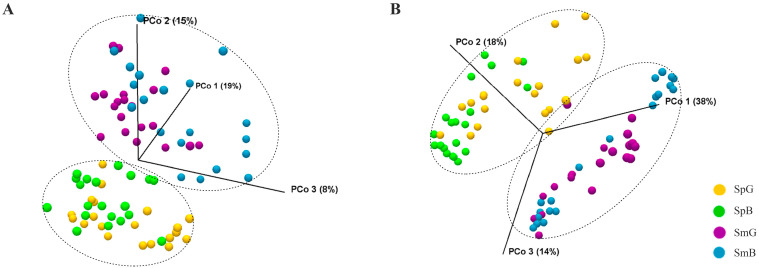
UniFrac Principal Coordinate Analysis (PCoA) plot. PCoA plots depicting the (**A**) unweighted UniFrac (quantitative) and (**B**) weighted UniFrac (qualitative) were implemented to illustrate the β-diversity. Each figure indicates the microbiota of chicory leaf samples collected during each season and at each site (SpG; Gongju sample collected in spring, SpB; Busan sample collected in spring, SmG: Gongju sample collected in summer, and SmB; Busan sample collected in summer).

**Figure 2 microorganisms-11-01560-f002:**
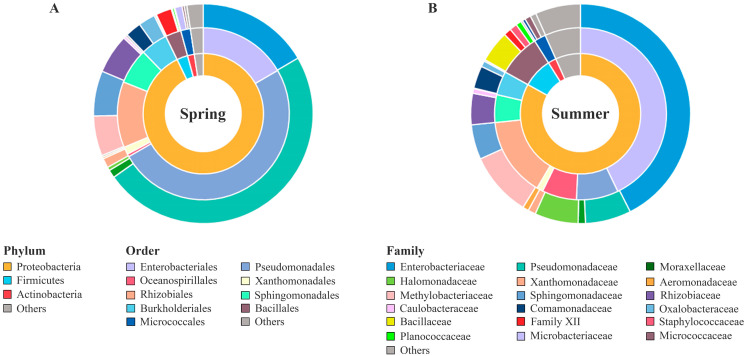
Comparison of microbiota composition at the phylum, order, and family levels in (**A**) spring and (**B**) summer. ‘Others’ indicate microbial phyla, order, and families with relative abundance below 2%, 1%, and 0.5% in the sample average, respectively.

**Figure 3 microorganisms-11-01560-f003:**
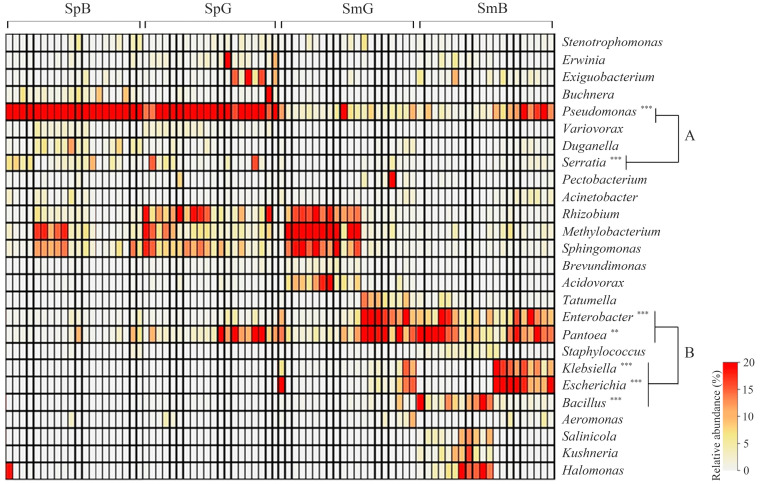
Analysis of microbiota composition at the genus level. Heatmap analysis shows the genus level relative abundance (more than average 0.5%) on chicory samples. Samples were clustered by Spearman’s rank correlation (**, *p* < 0.01 and ***, *p* < 0.001).

**Figure 4 microorganisms-11-01560-f004:**
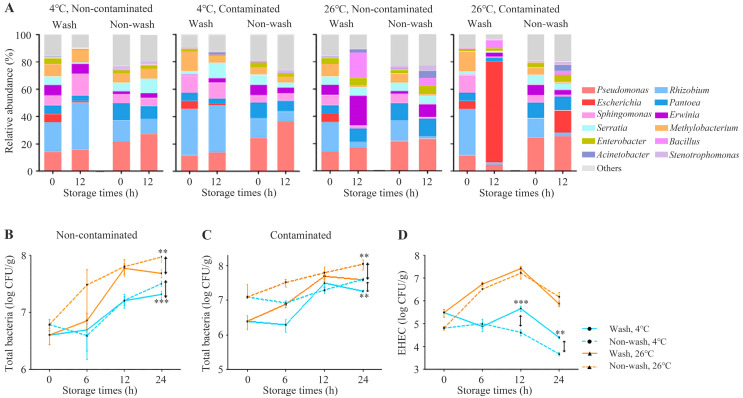
Change of chicory microbiota depending on various storage conditions (contamination, wash, and time). (**A**) Shifts in microbiota composition at the genus level of chicory following experimental contamination with EHEC and storage under different temperatures and washing conditions. The genera with an average relative abundance of less than 1% of each sample were indicated as ‘Others’ (grey-colored bar). The number of total bacteria and EHEC were quantified in chicory stored at 4 °C and 26 °C after washing and unwashing treatment. Amounts of total bacterial loads in (**B**) non-contaminated (**C**) contaminated samples over time. (**D**) EHEC (*E. coli* ATCC 35150) loads were quantified under different storage conditions (**, *p* < 0.01 and ***, *p* < 0.001).

**Figure 5 microorganisms-11-01560-f005:**
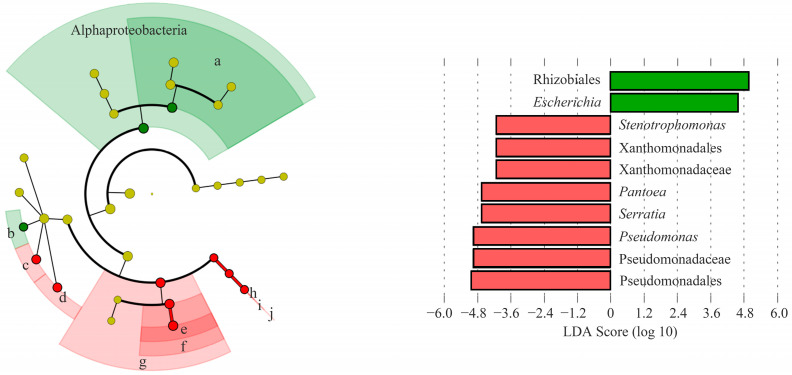
Cladogram and linear discriminant analysis (LDA) coupled with effect size measurement (LEfSe) analysis identified the most differentially abundant taxa that distinguish between wash and unwashing treatments. By LEfSe analysis based on the LDA score, differences are represented by the color of the most abundant genera in the non-contaminated chicory stored at 4 °C for 12 h (red; non-wash and green; wash group). The genera with both averages of relative abundance over 1% and LDA scores above 2 are shown (LDA score ≥ 2, *p* < 0.05). a: Rhizobiales, b: *Escherichia*, c: *Pantoea*, d: *Serratia*, e: *Pseudomonas*, f: Pseudomonadaceae, g: Pseudomonadales, h: *Stenotrophomonas*, i: Xanthomonadaceae, and j: Xanthomonadales.

**Table 1 microorganisms-11-01560-t001:** Summary of total bacterial loads and α-diversity indices of chicory obtained at each sampling time and site.

Sampling Information	Normalized Reads	Total Read/Sample	Total Bacteria (Average log CFU/g)	α-Diversity Indices		
				Chao1	Observed OTUs	Shannon
Spring	18,840	41,494	6.78 ^A^	610.09 ^A^	379.48 ^B^	4.51 ^B^
Summer		34,907	6.11 ^B^	600.3 ^A^	431.37 ^A^	5.17 ^A^
Gongju		39,105	6.76 ^a^	635.13 ^a^	418.72 ^a^	5.18 ^a^
Busan		37,296	6.19 ^b^	575.26 ^a^	392.12 ^b^	4.51 ^b^

-The seasonal bacterial counts (CFU/g) within the same site (Gongju and Busan, respectively) or the regional bacterial counts within the same season (spring and summer, respectively) were compared. -The same uppercase and lowercase letters (A, B and a, b) for each column were not significantly different by the Students’ *t*-test at *p* < 0.05. ^A, B^; Between sampling times. ^a, b^; Between sampling sites.

**Table 2 microorganisms-11-01560-t002:** Quantification of the potential pathogens in chicory samples through qRT-PCR.

Sampling Times	Bacterial Loads of Pathogenic Bacteria (CFU/g)							
	EHEC	EPEC	ETEC	*K. pneumoniae*	*B. cereus*	*S. marcescens*	*S. aureus*	*A. lwoffii*
Spring	6.59 × 10^2^ (5.00% *)	3.28 × 10^2^ (7.50%)	N.D. **	N.D.	5.12 × 10^3^ (2.50%)	N.D.	N.D.	N.D.
Summer	N.D.	N.D.	N.D.	7.36 × 10^2^ (8.75%)	N.D.	2.57 × 10^3^ (1.25%)	N.D.	1.09 × 10^2^ (2.50%)

* The average detection rate in 40 samples. ** Non-detected.

## Data Availability

Data will be made available on request.

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
