# Peer review of "Characterization of the Bacterial Communities in Cichorium intybus According to Cultivation and Storage Conditions"

_microorganisms, 2023, doi:10.3390/microorganisms11061560_

Round 1

Reviewer 1 Report

See attached file

A few grammatical issues and some (non-language related) incomplete of confusing sentences were found. Maybe having the manuscript proof-read by a native english-sqpeaking colleague or professional would help.

Author Response

We would like to thank you for your sincere review. Please see the attachment.

Reviewer 2 Report

Bacterial name needs to be Italic.

Only Spring and Sumer, this design, is not accurate, and suggest to add normal temp info.

Bioinformatical analysis is typical and normal, and needs more explanation based on this study objective.

Author Response

(The authors gave the same response as above.)

Round 2

Reviewer 1 Report

Second Review Report

Microbiota dynamics in Cichorium intybus (chicory) leaves under different storage conditions, now: Characterization of the bacterial communities in Cichorium intybus according to cultivation and storage conditions. 

Microorganisms-2428626-V2

This manuscript was improved considerably, just minor issues remained to be addressed. These issues seem to be easy-to-resolve if the authors go back to the original data and confirm the data analysis.  Suggested revisions are listed below:

Table 1:

1.     In their comments, the authors indicate that there was no difference in bacterial counts between spring and summer for each of the regions sampled. However, in Version 1, Table 1 showed counts that were consistently lower in the summer compared to spring in both regions. The standard deviation may have determined this. This is one of the various reasons why this reviewer suggested that the counts be transformed into log values. This reviewer acknoledges that, although log transformation for bacterial count data is proven to be useful when counts are obtained from MPN or plate counts to allow parametric test such as t-test, it is unknown whether counts obtained by qPCR have a normal distribution and do not need to be transformed. Therefore, this reviewer continues to suggest at least a verification of data normality in SAS, or data transformation to compare statistical differences (or lack thereof) between seasons for each region group. 

2.     If the lack of difference is confirmed, Table 1 is adeqaute as it is not, but the authors will need to add a line in the text to clarify that there was no difference. This will ensure that readers understand why data are pooled in the table.

References:

Remove double space in the references or double-space the remaining of the manuscript. For this, please follow the journal’s guidelines.

Other, line-specific comments:

Ln 26. Produce is a non-countable noun, therefore there is no plural or singular for the term produce. Please add simply produce (no final s). If plural is preferred, then say something like produce items, or another form that is grammatically acceptable.      

Author Response

Thank you for the review. Please see the attachment file for revision #2. 
